# Systematic review on cost-effectiveness analysis of school-based oral health promotion program

Thinni Nurul Rochmah[1,2☯]*, Aulia Ramadhani[3☯], Taufan Bramantoro[3☯], Lucindari Gea Permata[3,4☯], Tin Zar Tun[5☯]

1 Department of Health Policy and Administration, Faculty of Public Health, Universitas Airlangga, Surabaya, Indonesia, 2 The Airlangga Centre for Health Policy Research Group, Surabaya, Indonesia, 3 Department of Dental Public Health, Faculty of Dental Medicine, Universitas Airlangga, Surabaya, Indonesia, 4 Graduate Program of Dental Health Science, Faculty of Dental Medicine, Universitas Airlangga, Surabaya, Indonesia, 5 Department of Paediatric Dentistry, University of Dental Medicine, Yangon, Myanmar

☯ These authors contributed equally to this work.
* thinni_nurul@fkm.unair.ac.id

## Abstract

### Objective

This study aims to assess the variance of the cost-effectiveness ratio of the school-based oral health promotion and prevention program for elementary school children.

### Methods

This review protocol was registered in the international database of Prospectively Registered Systematic Reviews in Health and Social Care (PROSPERO) (No: CRD 42022326734). The search for articles conducted in March-April 2022 focuses on any kind of school-based Promotive and Preventive Program for elementary school children that have control groups, and the outcome was Incremental Cost-Effectiveness Ratio (ICER). Grey literatures are not eligible. This review used five databases (PubMed, Scopus, Web of Science, CINAHL, and Google Scholar). Two independent reviewers referred to the PICO for inclusion and exclusion criteria and carried out the systematic review process. JBI ACTUARI Guidance for Critical Appraisal of Economic Evaluation Assessment Tools was used to assess the quality of the study.

### Results

Of the total 1,473 articles found, there were 5 articles that matched the article search criteria and were included in a systematic review. It was known that the labor cost has a large proportion of the total program cost, and cost-saving programs were found in the two milk fluoridation programs (18.59 USD and 1.7 USD/averted DMF-S), fluoride mouth rinsing program (10.86 USD), and a comprehensive program with glass ionomer cement (461,766.37 USD/ averted DALYs).

**Data Availability Statement:** All relevant data are within the paper and its Supporting information files.

**Funding:** The author(s) received no specific funding for this work.

**Competing interests:** The authors have declared that no competing interests exist.

## Conclusion

The fluoride programs and the comprehensive program with glass ionomer cement have the lowest cost-effectiveness ratio.

## Introduction

According to data from The Global Burden of Disease Study in 2017, approximately 3.5 billion people worldwide experience dental and oral health problems, with the highest prevalence among elementary school-aged children [1, 2]. This is supported by a recorded 0.5 billion cases of dental caries in children aged 0 to 14 years in 2019, making it the disease with the largest prevalence in this age group [3]. The dental caries were found in both primary and permanent teeth [4]. Research conducted by Li and Wang in 2002 supports the evidence that caries status in permanent teeth is closely related to the status of primary teeth [5]. While not life-threatening, oral diseases have been ranked as the fourth most expensive disease to treat in most industrialized countries and account for 5–10% of public health expenditure is used for oral health care in high-income industrialized countries [6].

Globally, the direct cost of treating dental and oral diseases was $356,770,000,000, with the United States spending the most at $119,070,000,000. The indirect cost in the form of productivity loss was $187,610,000,000. Thus, the overall expense amounted to $544,410,000,000 [7]. Expenditures for the treatment of dental and oral diseases are expected to continue increasing. According to research conducted by Jevdjevic in 2021, his team created a prediction model for calculating dental health expenditure in 32 countries worldwide, which reached $316.550 million in 2020, $425.180 million in 2030, and $593,410,000,000 in 2040 [8].

Given the significant economic losses caused by dental caries, dental care costs can pose a substantial health and economic burden, particularly for children in low and middle-income countries. The most effective approach to reduce this burden is to prioritize preventive and promotive measures [9, 10]. Overcoming this issue, the school-based oral health promotion and prevention program for school children has been implemented by 72.4% of countries worldwide. However, despite over 50% of countries implementing such programs for children's dental health, the prevalence of dental caries remains high. The clinical and cost-effectiveness of these programs have not been evaluated.

Cost Effectiveness Analysis (CEA) evaluates the costs and consequences of alternative interventions using clinical outcomes in "natural units". CEA is used to answer questions about productive or production efficiency. The analysis focuses on the cost and effectivity differences between two or more treatment options. Therefore, the analysis performed tends to refer to incremental costs, incremental effects, and Incremental Cost-Effectiveness ratio (ICER) [11].

Based on the World Health Organization's (WHO) Global Health Expenditure data from 2016 to 2019, the percentage of health expenditure in the promotive and preventive sectors is significantly lower compared to the curative sector [12]. The low health expenditure in the promotive and preventive sectors highlights the potential for maximizing efforts in prevention and health promotion.

To date, there is no research or review suggesting the most effective school-based promotive and preventive programs for elementary school children in terms of cost and effectiveness. Thus, the aim of this study is to conduct a systematic evaluation to analyze the cost-effectiveness of the school-based oral health promotion and prevention program for school children.

## Methods

The systematic review protocol was registered in the international database of Prospectively Registered Systematic Reviews in Health and Social Care (PROSPERO) (No: CRD 42022326734). The research question applied to this study was "What kind of school-based oral health promotive and preventive program targeted elementary school children are categorized as the cost-saving program?".

### Eligibility criteria

This study included any kind of original primary research (intervention or observational studies) articles using cost-effectiveness analysis that was published in peer-reviewed journals between January 1990 and March 2022. Study that compared several oral health programs or caries prevention activities in one paper are also included. For the inclusion criteria, PICO were used as follows:

- Population of interest (P) was elementary school children aged 5 to 12 years old [13].

- Intervention (I) or exposure (E) of interest were school-based oral health promotive and preventive programs, which typically include children's dental check-ups, joint tooth brushing activities, dental and oral health education activities, nutritious food management activities, and dental caries prevention care. Intervention activities also include primary prevention, secondary prevention, and tertiary prevention activities. Both medical and non-medical personnel can carry out the treatments and activities.

- Comparator or control (C) group consisted of children who were not exposed to the promotive and preventive programs, or were given different programs than those mentioned in the intervention/exposure.

- Outcome (O) was the Incremental Cost-Effectiveness Ratio (ICER), which compares the incremental cost and incremental effects of the programs.

### Search strategy

The literature search was carried out between March and April 2022 and included a search for secondary data from peer-reviewed international journals using predetermined keywords based on PICO terms. The keywords used included "Elementary school children", "School aged children", "Elementary aged", "Primary school", "Preventive health services", "School health program", "Dental caries prevention", "Oral health promotion program", "Oral health care", "Oral health", "Cost effectiveness analysis", "Incremental cost", "Incremental effects", "ICER", and "Dental caries".

To ensure comprehensive coverage, the literature search was conducted in five recommended database for systematic reviews in the health sector [14–16]. This study used the following databases: PubMed (searched on March, 2022), Scopus (searched on March, 2022), Web of Science (searched on March,2022), CINAHL (searched on April, 2022), and Google Scholar (searched on April, 2022).

### Study selection

After duplications were removed, all articles were chosen in two steps. Step 1: Using PICO criteria, the study was chosen based on its title and abstract. Step 2: Conduct full-text screening on the eligible articles and selecting those that met the criteria for inclusion. Study selection results are reported in the PRISMA flow chart. PRISMA flow charts depict

the flow of information through the various phases of a systematic review. The study selection reports the results of mapping the number of records identified, included and excluded, as well as the reasons for exclusion [17]. Study selection was carried out by two independent reviewers.

## Quality assessment method

The study quality assessment was carried out using the JBI ACTUARI Guidance for Critical Appraisal of Economic Evaluation Assessment Tools [18]. The assessment of the quality of the study was carried out by two independent reviewers, each of whom made an assessment based on a predetermined checklist with the answers "Yes", "No", and "Unclear". The results of the final assessment are discussed together. To assess the study limitations and applicability, the NICE-GRADE checklist was used [19].

## Data extraction

The included articles were extracted based on several criteria including country of origin, type of study, year of cost calculations, study perspective, type of health economic evaluation reporting guidelines, population, intervention/exposure, comparator, outcome, currency, intervention cost (total cost of the program), control cost (cost of the compared program or treatment cost that patient have to pay), incremental cost (intervention cost minus control cost), the effect of intervention (changes in oral health status due to intervention), effect of control (changes in oral health status due to compared program or current oral health status without any intervention), incremental effect (effect of intervention minus effect of control), and cost-effectiveness ratio (incremental cost per incremental effects). These data extraction were based on research questions or objectives, study designs, and outcomes that have been previously established [20]. The extraction process also took into account the results of a previous systematic review [17]. Two independent reviewers carried out the data extraction process.

The data extracted from the articles included measurements that were originally reported by the original researchers and no recalculation were performed to determine the cost-effectiveness ratio or the total cost. If the original researcher did not provide data for a particular variable, the corresponding column in the data extraction table was left blank.

## Conversion of the currency

The currency conversion output utilized United States Dollars (USD) as the standard currency, which is widely accepted [21]. The cost data were converted to USD 2022 using the Campbell and Cochrane Economics Methods Group Evidence for Policy and Practice Information and Coordination Center (CCEMG-EPPI Center) cost converter [18].

## Data analysis

Data analysis was carried out in the form of a narrative synthesis of the results presented using Cost-Effectiveness Analysis (CEA) in accordance with the guidelines from The Joanna Briggs Institute (JBI) Reviewer's Manual for Systematic Review of Economic Evaluations [22].

## Data management and reporting

Data were managed and analyzed using Microsoft Excel and Zotero. The result of the literature search is presented according to the PRISMA flowchart and checklist.

## Results

The study selection process is presented in the PRISMA Flowchart (Fig 1). A total of 1,473 articles were identified in the identification stage, with 1,154 articles (78.3%) obtained from the PubMed database. After removing 139 duplicated articles, the remaining eligible articles underwent screening based on PICO criteria. To ensure research validity and reliability, the literature search was focused on peer-reviewed journals [23].

The first screening stage excluded 1,252 articles that did not meet the PICO criteria. Subsequently, 77 articles were further screened for full text. At the final stage of screening, five studies that met the PICO criteria were included in the analysis. Although Neidell et al. was initially included, it was later excluded due to the Average Cost-Effectiveness Ratio (ACER) outcome [24]. The analysis identified eight programs from the five included studies. The list of excluded studies can be seen in S1 Table.

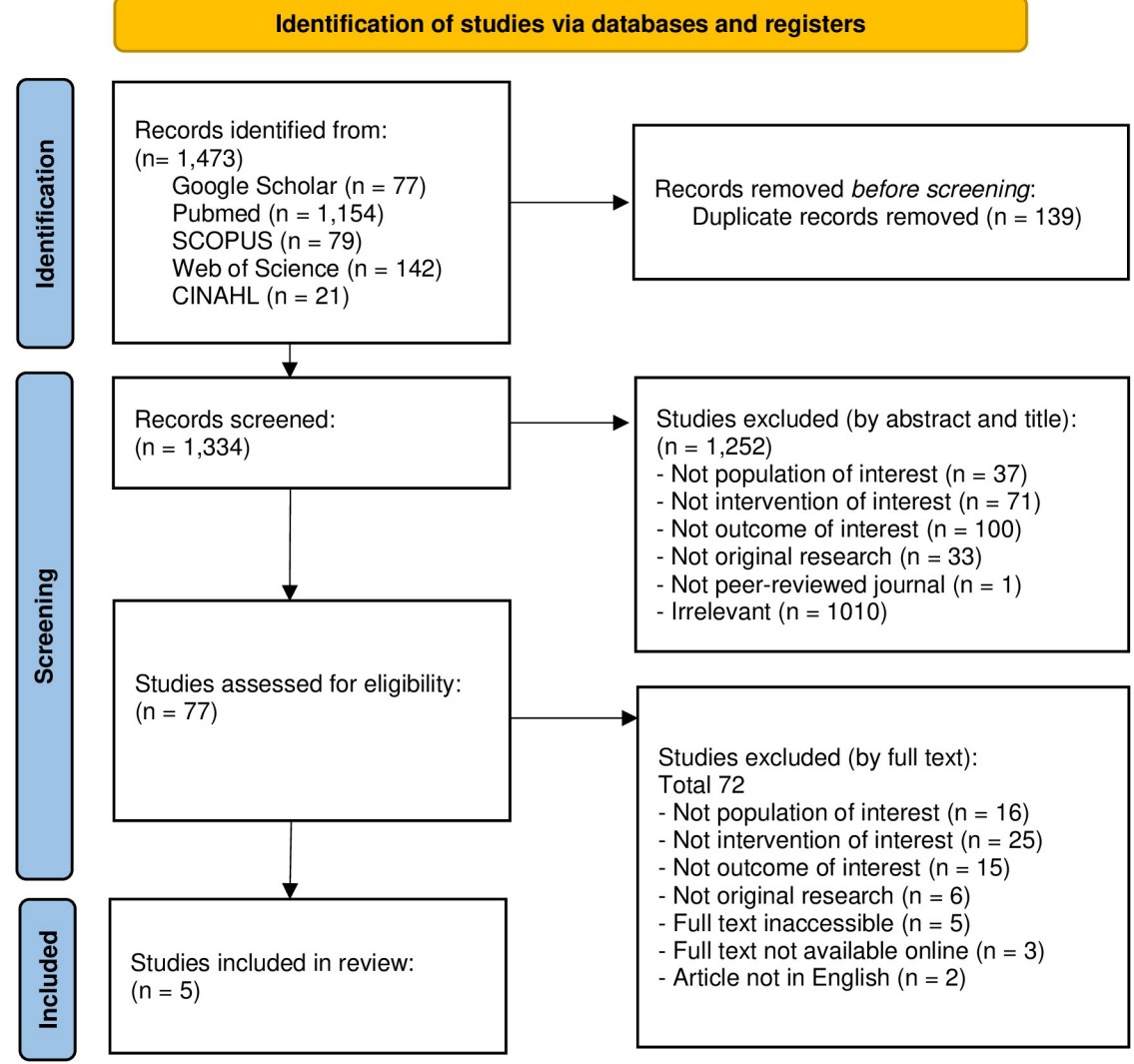

**Fig 1. PRISMA flowchart of the articles search process.**

Table 1. Quality assessment result of included studies.

| No | Scoring Indicators | Study | | | | |
|---|---|---|---|---|---|---|
| | | Lukssamijarulkul et al, 2022 | Tianviwat et al, 2020 | Marino et al, 2012 | Marino et al, 2018 | Huang et al, 2019 |
| 1 | Is there a well-defined question? | Yes | Yes | Yes | Yes | Yes |
| 2 | Is there comprehensive description of alternatives? | Yes | Yes | Yes | Yes | Yes |
| 3 | Are all important and relevant costs and outcomes for each alternative identified? | Yes | Yes | Yes | Yes | Yes |
| 4 | Has clinical effectiveness been established? | Yes | Yes | Yes | Yes | Yes |
| 5 | Are costs and outcomes measured accurately? | Yes | Yes | Yes | Yes | Yes |
| 6 | Are costs and outcomes valued credibly? | Yes | Yes | Yes | Yes | Yes |
| 7 | Are costs and outcomes adjusted for differential timing? | No | No | Yes | Yes | Yes |
| 8 | Is there an incremental analysis of costs and consequences? | Yes | Yes | Yes | Yes | Yes |
| 9 | Were sensitivity analyses conducted to investigate uncertainty in estimates of cost or consequences? | No | Yes | Yes | Yes | Yes |
| 10 | Do study results include all issues of concern to users? | Yes | Yes | Yes | Yes | Yes |
| 11 | Are the results generalizable to the setting of interest in the review? | Yes | Yes | Yes | Yes | Yes |
| | **Overall appraisal**[a] | Very serious limitation, partially applicable | Minor limitation, partially applicable | Minor limitation, partially applicable | Minor limitation, partially applicable | Minor limitation, directly applicable |

[a]Overall appraisal represented the final assessment results using JBI and NICE-GRADE checklist.

## Quality assessment results

The study quality assessment was carried out using the JBI ACTUARI Guidance for Critical Appraisal of Economic Evaluation Assessment Tool which includes 11 points of questions. The results of the assessment are discussed together to reach a joint decision on the assessment of the study [18]. The result can be seen in Table 1.

All included studies had well-designed research question, clearly identified alternative programs for comparison, identified cost and outcome components, the clinical effectiveness of each program, and include detailed outcome measurements or assessments on the research method. The studies were also evaluated using the NICE-GRADE scoring criteria to assess their limitations and applicability. Four studies were found to have minor limitations, while one study by Lukssamijarulkul et al. had a very serious limitation due to their failure to conduct the sensitivity analysis [25]. Sensitivity analysis is important in economic evaluation as it helps to assess the strength and robustness of the model's conclusions in the face of uncertainty [26].

## Characteristics of the study

Table 2 provides an overview of the characteristics of the included studies in terms of study design, economic evaluation perspective, health outcomes, currency, time horizon, and adherence to reporting guidelines. Among the five studies, four (57.1%) are model-based studies [27–30], and three (57.1%) used the societal perspective. All studies measured clinical effectiveness using the DMFT index, except for one study that used the Disability-Adjusted

**Table 2. Characteristic of the included studies.**

| Ref. No | Author (Year) | Country | Year of Cost Calculation | Study Design | Currency | Study Perspectives | Type of Intervention/ Exposure | Health Indicator | Time Horizon | Health Economic Guideline |
|---|---|---|---|---|---|---|---|---|---|---|
| 1 | Lukssamijarulkul et al (2022) | Thailand | 2015 | Trial Based | Baht | Healthcare providers | Comprehensive preventive and promotive program (including oral hygiene instructions, diet counselling, oral examination, fluoride varnish, and dental sealant) | DMF-T | 9 years | -[a] |
| 2 | Tianviwat et al (2020) | Thailand | 2020 | Model-based | Baht | Combination | Mobile dental service for dental sealant | Number of caries free teeth | 6 months | -[a] |
| 3 | Mariño et al (2012) | Chile | 2009 | Model-based | USD | Societal | 1. Milk fluoridation | DMF-T | 6 years | -[a] |
| | | | | | | | 2. Fluoride mouth rinsing program (FMR) | | | |
| | | | | | | | 3. Acidulated Phosphate Fluoride (APF) Gel | | | |
| | | | | | | | 4. Supervised Toothbrushing | | | |
| 4 | Mariño et al (2018) | Thailand | 2011 | Model-based | Baht | Societal | Fluoridated milk | DMF-S | 6 years | The Consolidated Health Economic Evolution Reporting Standard Guideline |
| 5 | Huang et al (2019) | United States of America | 2014 | Model-based | USD | Societal | Comprehensive oral health promotion program (including | DALYs | 5 years | -[a] |

[a]No health economic guidelines stated.

Life Years (DALYs) index. The studies used either USD (50%) or Baht (50%) in their cost calculations. The time horizon of the studies varied, with the longest being nine years [25] and the shortest being six months [27]. Only one study follow the health economic evaluation reporting guidelines.

## Program's cost analysis

All cost data have been adjusted to 2022 USD and are presented in Table 3. The cost provided in the data represents the calculation for one person. The most expensive intervention was mobile dental service for a dental sealant program, costing 20,944 USD 2022 for one patient [27]. Two studies did not clearly mention the intervention and control cost [28, 30].

One study calculated the total cost according to the number of teeth in both intervention and control groups [27]. The intervention group had a total of 2,647 teeth, while the control or comparison group had 1,474 teeth. In another study, the total costs calculated were program costs for 75,000 children, and the calculation of program costs per child was added below in the presented table [29]. The control cost was calculated from the cost of compared program or the dental treatment cost for treating dental problems that should be paid instead of implementing the intervention.

Four of the ten programs showed that the cost of the intervention program was lower than the cost of the compared program (control), which included two programs of milk fluoridation, fluoride mouth rinses, and a comprehensive program of promotive and preventive

**Table 3. Data extraction table of included studies.**

| Ref. No | Year of Cost Calculation | Population | Time Horizon | Type of Intervention/ Exposure | Comparator | Intervention Cost | Control Cost | Incremental Cost | Health Indicator | Effect of Intervention | Effect of Control | Incremental Effects | ICER |
|---|---|---|---|---|---|---|---|---|---|---|---|---|---|
| (1) | (2) | (3) | (4) | (5) | (6) | (7) | (8) | (9) = (7)-(8) | (10) | (11) | (12) | (13) = (11)-(12) | (14) |
| 1 | 2015 | Primary school students | 9 years | Comprehensive preventive and promotive program (including oral hygiene instructions, diet counselling, oral examination, fluoride varnish, and dental sealant) | Oral examination | 4,243.67 Baht (394.23 USD 2022) | 208 Baht (19.32 USD 2022) | 4,035.31 Baht (374.91 USD 2022) | DMFT | 1.00 | 2.00 | 1.00 | 4,035.31 Baht (374.9 USD 2022) / averted DMFT |
| 2 | 2020 | Primary school students (6–8 years old) | 6 months | Mobile dental service for dental sealant | Hospital-based dental clinic | 246,171 Baht (20,944 USD 2022) | 197,516 Baht (16,805 USD 2022) | 48,655 Baht (4,139 USD 2022) | Number of caries free teeth | 2.522 | 1.427 | 1.095 | 48,655 Baht (4,139 USD 2022) / averted dental caries |
| 3 | 2009 | Primary school students (12 years old) | 6 years | 1. Milk fluoridation | No intervention | -[a] | -[a] | -[a] | DMFT | 0.53 (average of effectivity) | -[a] | 0.53 (average of effectivity) | -14.78 USD (-18.59 USD 2022) averted DMFT |
|  |  |  |  | 2. Fluoride mouth rinsing program (FMR) | No intervention | -[a] | -[a] | -[a] | DMFT | 0.26 (average of effectivity) | -[a] | 0.26 (average of effectivity) | -8.63 USD (-10.85 USD 2022) / averted DMFT |
|  |  |  |  | 3. Acidulated Phosphate Fluoride (APF) Gel | No intervention | -[a] | -[a] | -[a] | DMFT | 0.21 (average of effectivity) | -[a] | 0.21 (average of effectivity) | 8.55 USD (10.75 USD 2022) / averted DMFT |
|  |  |  |  | 4. Supervised Toothbrushing | No intervention | -[a] | -[a] | -[a] | DMFT | 0.24 (average of effectivity) | -[a] | 0.24 (average of effectivity) | 21.30 USD (26.79 USD 2022) / averted DMFT |
| 4 | 2011 | Primary school students | 6 years | Fluoridated milk | No intervention | 432 Baht[b] (42.53 USD 2022) | 540.67 Baht[b] (53.23 USD 2022) | -108.67 Baht[b] (-10.70 USD 2022) | DMFS | 1.06 | 1.60 | 0.54 | - 109 Baht (-10.73 USD 2022) / averted DMFS |
| 5 | 2014 | Children age 5–12 years old | 5 years | Comprehensive oral health promotion program (including | No intervention | -[a] | -[a] | -239.77 USD (-307.02 USD 2022) | DALYs | -[a] | -[a] | 0.0006 | -400,645.52 USD (-461,766.37 USD 2022) / averted DALYs |

[a]The data was not clearly stated by authors.

dental health for children using glass ionomer materials to prevent the expansion of dental caries. These programs have negative incremental cost value.

According to Table 3, the comprehensive promotive and preventive dental health program with Glass Ionomer Cement as a sealant was a cost-saving program that showed a lowest incremental cost ((-) 307.02 USD) [30]. Additionally, most of the fluoride programs were also found to be cost-saving.

Overall, the intervention programs showed greater clinical effectiveness than the compared programs, with positive incremental effect values (Table 3). Most of the programs were compared to a no-intervention condition. Despite there being some variations in health indicators used in each included study, most studies utilized dental caries indicators such as the DMF-T index, the number of caries-free teeth, the percentage of the program's effectiveness (ICER measurement used DMF-T), and DMF-S index.

The analysis results indicate that four programs (milk fluoridation and fluoridated mouth rinses [28], fluoridated milk [29], and comprehensive oral health promotion program using glass ionomer sealant [30]) are favorable for intervention due to their negative incremental cost and positive incremental effects. On the other hand, the remaining four programs (Comprehensive oral health preventive and promotive program [25], APF-Gel and Supervised toothbrushing [28], Mobile dental service for dental sealant [27]) were deemed "unclear" and required further judgment to determine whether intervention was preferable, considering incremental cost-effectiveness measures and priorities or willingness to pay.

## Discussion

Five studies comprising eight oral health promotive and preventive programs for primary school students, ranging in age from six to twelve years old population, were assessed. Three of the studies were conducted in primary schools in Thailand, one in Chile, and another one in The United States of America. In 2002, WHO recommended school-based oral health promotion interventions due to schools' capacity to reach all school-age students and facilitate relationships with parents and the community [31]. Despite such recommendation, the situation may differ in each schools in each continents, which need to be considered. Within the included studies, no specific condition or oral health policy were mentioned that could interfere with the clinical effectiveness results for each school in different countries. Additionally, differences in continents led to variations in setting between the studies, such as program's objectives, delivery model, target population, funding, school's systems, and support. The unique needs, environments, and resources of every nation can be reflected in the broad range of school-based oral health promotion initiatives.

Based on the assessment results, all interventions showed positive clinical effectiveness compared to the non-intervention group. However, all fluoride programs were found to be cost-effective, as they were cost saving and demonstrated good clinical effectiveness. This finding is consistent with a previous study conducted by Davidson [32]. Fluoride treatment has been shown to be an effective intervention in preventing dental caries. However, the implementation of fluoride treatment may differ across countries depending on their available resources and infrastructure. Despite these differences, the clinical and cost effectiveness of fluoride programs remain valuable for public health initiatives aimed at improving oral health and reducing disparities in dental health outcomes. It is important to consider the feasibility of implementing fluoride programs in various settings and to adapt delivery strategies accordingly to maximize their impact.

One of the key principles of interpreting CEA is considering the analytic perspective [33]. Most of the studies included in the assessment used a societal-perspective, which includes cost

components from the patients perspectives, such as time spent on the way to the dental facility or to get treatment. This approach was applied in model-based studies that compared the clinical effectiveness of the proposed program with no-intervention conditions, where the cost of control was the treatment cost for dental caries, leading to a higher total cost compared to the proposed program's cost. This might affect the interpretation of ICER comparison between model-based studies and trial-based studies.

Another key finding from the assessment was that labor costs were a significant component that increased the program's costs. This finding is consistent with a previous study on conventional fissure sealant program [24]. For programs that provide direct oral health services, such as dental care or oral health promotion and education initiatives, labor expenses are likely to represent a substantial portion of the budget. Cost for dental professionals' salaries and benefits, as well as the expenses of hiring and training support staff, can added up quickly. The sbudget for oral health programs may include a significant portion of labor expenses, although the exact percentage may vary depending on the services provided and the program's location, among other things.

To determine which program is worth implementing in a particular area, it is necessary to establish the cost-effectiveness threshold or willingness to pay rate. Although most of the studies did not mention the threshold, the authors considered several factors to determine the cost-effectiveness of these programs. However, implementation decisions should not solely rely on cost-effectiveness evidence, but also considering the situations and needs of the target population [34, 35]. Additionally, dental-related outcomes pose constraints on policymakers when allocating budgets for health programs. The use of DALYs increases the usefulness of economic evaluations to allocate funds for the dental health sector from the general health assessment [30]. Therefore, the use of the DALYs as an indicator is preferable for health policymakers. The results of this study are expected to provide an overview for designing oral health intervention programs that can be implemented in communities or populations, particularly in middle and low-income countries, where caries and dental health issues are still prevalent despite limited resources. When budgeting for oral health programs, all expenses must be considered, and activities must be prioritized based on how they might affect the outcomes of oral health.

## Limitations of the study

There are several limitations of this study. Firstly, there is heterogeneity in the health indicators used in each article, which makes it difficult to compare the clinical effectiveness across all studies. Additionally, most of the studies used model-based, which may limit the generalizability of the study. Furthermore, the majority of the studies were classified as having a minor limitation and partially applicable, and only one study followed the guidelines for reporting health economic evaluation. Another limitation is that the study was limited to primary school-based setting, and it was challenging to determine the criteria for school-based term and resulting in the missing of potential studies.

## Conclusion

Oral Health Prevention and Promotion Programs planning should consider several factors, including the total cost of the program, clinical effectiveness, and cost-effectiveness between the proposed program and current program or situations. Among the program costs, labor cost was found to be the most influential. However, reducing program's cost by having various human resource options, such as dental assistant who can substitute for dentist. In the current

study, the milk fluoridation program, Fluoride Mouth Rinses (FMR), and a comprehensive program using glass ionomer were found to be cost-saving programs.

## Supporting information

**S1 Table. Excluded studies list.**
(PDF)

**S2 Table. PRISMA 2020 for abstract checklist.**
(PDF)

**S3 Table. PRISMA 2020 checklist.**
(PDF)

## Acknowledgments

The authors would like to thank Dr. Ernawaty (Faculty of Public Health, Universitas Airlangga, Surabaya), Dr. Elida Zairina (Faculty of Pharmacy, Universitas Airlangga, Surabaya), Dr. Anon, and Dr. Gerryd (Surabaya Public Health Center) for their reviews and suggestion for this manuscript.

## Author Contributions

**Conceptualization:** Thinni Nurul Rochmah, Aulia Ramadhani.

**Data curation:** Aulia Ramadhani, Lucindari Gea Permata.

**Formal analysis:** Thinni Nurul Rochmah, Aulia Ramadhani.

**Investigation:** Thinni Nurul Rochmah, Aulia Ramadhani.

**Methodology:** Thinni Nurul Rochmah, Aulia Ramadhani, Taufan Bramantoro.

**Project administration:** Aulia Ramadhani.

**Resources:** Aulia Ramadhani.

**Software:** Aulia Ramadhani.

**Supervision:** Thinni Nurul Rochmah, Taufan Bramantoro.

**Validation:** Thinni Nurul Rochmah.

**Writing – original draft:** Thinni Nurul Rochmah, Aulia Ramadhani, Taufan Bramantoro, Lucindari Gea Permata.

**Writing – review & editing:** Thinni Nurul Rochmah, Aulia Ramadhani, Taufan Bramantoro, Lucindari Gea Permata, Tin Zar Tun.

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
