## [Decision Letter · Decision Letter 0]

15 Feb 2023

PONE-D-23-00755Cost-Effectiveness Analysis of school-based oral health promotive and preventive program for elementary school children: A systematic reviewPLOS ONE

Dear Dr.Thinni Nurul Rochmah

Thank you for submitting your manuscript to PLOS ONE. After careful consideration, we feel that it has merit but does not fully meet PLOS ONE’s publication criteria as it currently stands. Therefore, we invite you to submit a revised version of the manuscript that addresses the points raised during the review process.

We look forward to receiving your revised manuscript.

Kind regards,

Fahad Umer

Academic Editor

PLOS ONE

Journal Requirements:

Reviewers' comments:

Reviewer's Responses to Questions

**Comments to the Author**

1. Is the manuscript technically sound, and do the data support the conclusions?

Reviewer #1: Yes

Reviewer #2: Yes

2. Has the statistical analysis been performed appropriately and rigorously? 

Reviewer #1: N/A

Reviewer #2: Yes

3. Have the authors made all data underlying the findings in their manuscript fully available?

Reviewer #1: Yes

Reviewer #2: Yes

4. Is the manuscript presented in an intelligible fashion and written in standard English?

Reviewer #1: No

Reviewer #2: No

5. Review Comments to the Author

Reviewer #1: This is a systematic review of cost-effectiveness studies of school-based oral health programs for elementary school children. It found 5 studies (consisting of 8 programs) out of which 4 programs were found to be cost saving and the other 4 potentially cost-effective.

The study is registered in Prospero and generally follows the PRISMA-standard. However, I found some weaknesses with it and have some comments.

The title is misleading, you do not do a CEA, you do a SR on other CEAs.

In several parts of the study, the English language needs to be improved.

The setting of the oral health program is not presented or discussed in detail, even though that is highly influential when performing promotive and preventive programs. School systems in different countries tend to differ.

Negative cost-effectiveness ratios should not be presented in values as such ratios can’t be interpreted (more than saying that it is dominant). Similar, the use of terms such as “most cost-effective” is not correct, which you use in your research question.

You state that you assess the risk of bias, but I can’t find any presentation of this. Your study quality assessment is more of an indicator whether certain aspects are covered in the analysis, rather than analysing the risk-of-bias of those aspects within the study.

Some parts of the result sections belong to the discussion.

I was surprised that only five studies were included. I think you need to present the reference to the 72 excluded full text studies in appendix and present what main criteria they failed to fulfil.

In my opinion the use of the dominance ranking matrix analysis does not provide any additional valuable information in this systematic review. Hence, I think Table 4 and 5 can be omitted, this information is already presented in Table 3.

There are unnecessary part in the discussion, you don’t have to explain what a CEA and an ICER is or how to interpret it, that should (if needed) have been presented in the introduction. However, you need to compare your findings with other studies. There are several systematic reviews in child oral health programs that should have been discussed here (even though they have not used school as a strict criterion for the programs), for example by Hettiarachchi et al 2018, Fraihat et al 2019, Eow et al 2019, Davidson et al 2021.

You write that there are several limitations with the study, but you only mention two such ones (variety of health indicators and your searching keywords). I think there are other more important weaknesses such as lack of risk-of-bias-analysis, presentation of the settings, potential missing studies etc.

Reviewer #2: 

Thank you for addressing a very important topic for the economic evaluation of different preventive and oral health promotion programs in elementary school-going children. The manuscript is adequately written however, there is some room for improvement.

Attached below are the comments for your reference:

**Major points:**

The manuscript is quite wordy and should be significantly reduced in length.There are several formatting errors, especially the referencing. References should be cited at the end of the statement and after full stop.Try to write the manuscript in third-person past tense and get it reviewed by someone proficient in English writing.

**Introduction:**

Very well written; writing is supported by evidence.Line 48-51: There is redundancy, can be rephrased to a single statement.Line 64: Reference should never be in the middle of the statement.Some details about cost-effectiveness analysis and its utility can be added here in the introduction.

**Methods:**

Fairly adequate description of the methodology; clear research question and PICOs model.The search terms are inadequate. Should have included more terms like incremental cost, incremental effects, ICER, QALYs, DALYs etc.Line 116-118: This information is not needed.Line 173-174: The authors have used JBI checklist for systematic review of economic evaluations however, they should have also used the GRADE tool to determine the certainty in evidence.

**Results:**

Line 229-230: Amongst the included studies, only a single study has reported outcome in the form of DALYs which is an outcome measure for cost-utility analysis. The authors could have excluded this study to reduce heterogeneity. Else they could have expanded their search to include QALYs, DALYs, and cost-utility analysis along with some other relevant terms to yield more comprehensive results.Table 3 provides all the information regarding cost of interventions as well as control, however, there is no clear information in text as to what exactly the control group was because there is some cost that is mentioned for control group as well in table 3.The authors did not report whether the included studies were reported according to health economics guidelines e.g., CHEERS checklist.

**Discussion:**

Too lengthy discussionFindings should be summarized, too much extra information.Line 302-314: Too wordy, can be made brief.Line 322-330: Too much information about a cost-effectiveness analysis, can be removed from here and included in the introduction.Line 427-432: Not required.

**Limitations:**

The authors could have mentioned another limitation as the majority of included studies were model-based hence the results could not be adequately generalizable; this is an inherent limitation of cost-effectiveness analysis based on analytic modeling approach as the data on which the model is built may be biased in some way or the other thus, limiting the generalizability of the findings.**********

6. PLOS authors have the option to publish the peer review history of their article (what does this mean?). If published, this will include your full peer review and any attached files.

Reviewer #1: No

Reviewer #2: **Yes: **Nighat Naved

---

## [Author Response · Author response to Decision Letter 0]

28 Mar 2023

Reviewer #1

1. The title is misleading, you do not do a CEA, you do a SR on other CEA’s

= The title changed to “Systematic review on Cost-Effectiveness Analysis of School-Based Oral Health Promotion Program (Page 1 / Line 1)

2. In several parts of the study, the English language needs to be improved

= The manuscript has been sent to the proofreading company for an English brush-up.

3. The setting of the oral health program is not presented or discussed in detail, even though that is highly influential when performing promotive and preventive programs. School systems in different countries tend to differ.

= Added in discussion section (Page 22)

4. Negative cost-effectiveness ratios should not be presented in values as such ratios can’t be interpreted (more than saying that it is dominant). Similar, the use of terms such as “most-effective” is not correct, which you use in your research question.

= All negative cost-effectiveness ratio has been reworded into cost-saving tendency. Research questions have been changed into “What kind of school-based oral health promotive and preventive program targeted elementary school children are categorized as the cost-saving program? (Page 2 & 14 / Line 36 & 256 - Page 4 / Line 84)

5. You state that you assess the risk of bias, but I can’t find any presentation of this. Your study quality assessment is more of an indicator whether certain aspects are covered in the analysis, rather than analysing the risk-of-bias of those aspects within the study.

= Limitation and applicability assessment added. (Page 10-11)

6. Some parts of the result sections belong to the discussion.

= Revised 

7. I was surprised that only five studies were included. I think you need to present the reference to the 72 excluded full text studies in appendix and present what main criteria they failed to fulfil.

= Appendix 1 attached.

8. In my opinion the use of the dominance ranking matrix analysis does not provide any additional valuable information in this systematic review. Hence, I think Table 4 and 5 can be omitted, this information is already presented in Table 3.

= Table 4 and 5 were omitted

9. There are unnecessary part in the discussion, you don’t have to explain what a CEA and an ICER is or how to interpret it, that should (if needed) have been presented in the introduction. However, you need to compare your findings with other studies. There are several systematic reviews in child oral health programs that should have been discussed here (even though they have not used school as a strict criterion for the programs), for example by Hettiarachchi et al 2018, Frahait et al 2019, Eow et al 2019, Davidson et al 2021.

= Discussion revised (Page 22)

10. You write that there are several limitations with the study, but you only mention two such ones (variety of health indicators and your searching keywords). I think there are other important weakness such as lack of risk-of-bias analysis, presentation of the settings, potential missing studies etc.

= Revised. Generalizability, studies’ risk of bias, the use of health economic guidelines, and missing of potential studies were added as study limitations. (Page 30)

 

Reviewer #2

Major points:

1. The manuscript is quite wordy and should be significantly reduced in length.

= Word reduced.

2. There are several formatting errors, especially the referencing. References should be cited at the end of the statement and after full stop. 

= Revised. 

3. Try to write the manuscript in third-person past tense and get it reviewed by someone proficient in English writing.

= Done 

Introduction

4. Very well written; writing is supported by evidence 

5. Line 48-51: There is redundancy, can be rephrased to a single statement.

= Line revised. (Page 3 / Line 51-52)

6. Line 64: Reference should never be in the middle of statement.

= Revised (Page 3 / Line 66)

7. Some details about cost-effectiveness analysis and its utility can be added here in the introduction.

= Added. (Page 4 / Line 69)

Methods

8. Fairly adequate description of the methodology; clear research question and PICOs model. 

9. The search terms are inadequate. Should have included more terms like incremental cost, incremental effects, ICER, QALYs, DALYs, etc.

= In the search keywords (for every database), we included “Incremental cost”, “Incremental effects”, and “ICER”. We did not include DALYs and QALYs to avoid getting more irrelevant articles. (Page 6)

10. Line 116-118: This information is not needed

= Information removed. (Page 6 / Line 120)

11 Line 173-174: The authors have used JBI checklist for systematic review of economic evaluations however, they should have also used the GRADE tool to determine the certainty in evidence.

= Limitation and applicability assessment added (using NICE-GRADE assessment tools) (Page 10-11)

Result

12. Line 229-230 Amongst the included studies, only a single study has reported outcomes in the form of DALYs which is an outcome measure for cost-utility analysis. The authors could have excluded this study to reduce heterogeneity. Else, they could have expanded their search to include QALYs, DALYs, and cost-utility analysis along with some other relevant terms to yield more comprehensive results. 

= We decided to include this study with the agreement of DALYs as a health indicator in mind that was needed to measure the incremental effect. Moreover, the study provides incremental cost, as well as incremental effect, that is needed to measure the cost-effectiveness ratio. 

13. Table 3 provides all the information regarding cost of interventions as well as control, however, there is no clear information in text as to what exactly the control group was because there is some cost that is mentioned for control group as well in table 3.

= The control group was explained in the eligibility criteria. Also in page 15 (Page 5 / Line 103)(Page 15/ Line 256)

14. The author did not report whether the included studies were reported according to health economic guidelines e.g., CHEERS checklist. 

= Information added in Table 2 (Page 13)

Discussion

15. Too lengthy discussion

= Word reduced 

16. Findings should be summarized, too much extra information.

= Revised

17. Line 302-314: Too wordy, can be made brief.

= Revised 

18. Line 322-330: Too much information about cost-effectiveness analysis, can be removed from here and included in the introduction.

= Revised 

19. Line 427-432: Not required.

= Omitted

Limitations

20. The authors could have mentioned another limitation as the majority of included studies were model-based hence the results could not be adequately generalizable; this is an inherent limitation of cost-effectiveness analysis based on analytic modelling approach as the data on which the model is built may be biased in some way or other thus, limiting the generalizability of the findings.

= Revised. Generalizability, studies’ risk of bias, the use of health economic guidelines, and missing of potential studies were added as study limitations (Page 30)

---

## [Editor Report · Decision Letter 1]

3 Apr 2023

Systematic review on Cost-Effectiveness Analysis of School-Based Oral Health Promotion Program

PONE-D-23-00755R1

Dear Dr.Thinni Nurul Rochmah ,

We’re pleased to inform you that your manuscript has been judged scientifically suitable for publication and will be formally accepted for publication once it meets all outstanding technical requirements.

Kind regards,

Fahad Umer

Academic Editor

PLOS ONE

---

## [Editor Report · Acceptance letter]

11 Apr 2023

PONE-D-23-00755R1 

Systematic review on Cost-Effectiveness Analysis of School-Based Oral Health Promotion Program 

Dear Dr. Rochmah:

I'm pleased to inform you that your manuscript has been deemed suitable for publication in PLOS ONE. Congratulations! Your manuscript is now with our production department. 

Kind regards, 

on behalf of

Dr. Fahad Umer 

Academic Editor

PLOS ONE